# Leaching Characteristics of Heavy Metals and Plant Nutrients in the Sewage Sludge Immobilized by Composite Phosphorus-Bearing Materials

**DOI:** 10.3390/ijerph16245159

**Published:** 2019-12-17

**Authors:** Shihe Li, Baihui Fang, Dongfang Wang, Xianqing Wang, Xiaobing Man, Xuan Zhang

**Affiliations:** 1College of Environmental Science and Engineering, Qilu University of Technology (Shandong Academy of Sciences), Ji’nan 250353, China; 2Jinan Zhongqian Environmental Protection and Energy Saving Technology Co., Ltd., Ji’nan 250100, China; 3Shandong Kanghui Technology Co., Ltd., Ji’nan 250101, China; 4Shandong Bluetown Anal & Testing Ltd., Ji’nan 250101, China

**Keywords:** sewage sludge, leaching characteristics, heavy metals, plant nutrients, cumulative release

## Abstract

In order to evaluate the environmental risk caused by land application of sewage sludge, leaching characteristics of heavy metals and plant nutrients in the sewage sludge immobilized by composite phosphorus-bearing materials were investigated. Their cumulative release characteristics were confirmed. Furthermore, the first-order kinetics equation, modified Elovich equation, double-constant equation, and parabolic equation were used to explore dynamic models of release. Results showed that sewage sludge addition significantly increased electricity conductivity (EC) in leachates, and the concentrations of heavy metals (Cu, Cr, Zn) and plant nutrients (N, P, K) were also obviously increased. The highest concentrations of Cu, Cr, and Zn in the leachates were all below the limit values of the fourth level in the Chinese national standard for groundwater quality (GB/T14848-2017). The immobilization of composite phosphorus-bearing materials reduced the release of Cu and Cr, while increased that of Zn. The fitting results of modified Elovich model and double-constant model were in good agreement with the leaching process of heavy metals and plant nutrients, indicating their release process in soil under simulated leaching conditions was not a simple first-order reaction, but a complex heterogeneous diffusion process controlled by multifactor.

## 1. Introduction

Sewage sludge is a residue produced during the biological wastewater treatment process. In China, its output has been increasing with the increasement of wastewater amount and treatment ratio. Sewage sludge is rich in organic matter (OM), nitrogen (N), phosphorus (P), and other trace elements such as Ca, Mg, Fe, Mo, B, etc [1,2]. Its land application can effectively utilize the useful resources and provide an important and low-cost alternative for sewage sludge disposal [3,4]. The properly treated sewage sludge is commonly used to improve soil quality [5,6]. Hamdi et al. conducted a field study over a three-year period under a semi-arid climate and found that repetitive sludge addition consistently improved total organic carbon (TOC), N, P, and K content up to soils treated with 120 t·ha^−1^·year^−1^, and impacted positively on biological properties, including microbial biomass and soil enzyme activities [7]. Tejada and Gonzalez indicated that land use of sewage sludge effectively reduced bulk density, aggregate instability, and soil loss under simulated rain at 140 mm·h^−1^ [8]. Cheng et al. suggested that sludge application increased soil cation exchange capacity (CEC), enhanced aggregate stability, and improved the ability of water and fertilizer conservation [9]. The sewage sludge amendment could result in robust plants with fast development and greater biomass production, by shortening their cultivation period [10,11,12].

However, sewage sludge can contain toxic heavy metals, such as Cu, Pb, Zn, Cd, Cr and so on. Its long-term land application would inevitably lead to accumulation of heavy metals in soil, posing a serious risk to surface water, groundwater, and even to human health [11,13,14]. Therefore, further studies are required to refine the migration and transformation of heavy metals before land use of sewage sludge [15]. Previous studies indicated that the mobility of heavy metals depends on the properties of soil [16,17], total concentrations of heavy metals and their speciation in sewage sludge [18,19], interaction of heavy metals with soil such as adsorption reaction [20], and complexation of heavy metals with organic or inorganic species [21,22].

Many kinds of chemical passivating agents have been used to immobilize heavy metals in sewage sludge and reduce the environmental risk of its land application [23]. The commonly used additives include basic compounds [24], aluminosilicates [25,26,27], phosphorus-bearing materials [28,29], and sulfides [30]. The long-term stability of the immobilized heavy metals under natural conditions has attracted wide attention [14]. Leaching in soil column was commonly used to simulate the migration and transformation of heavy metals in soil and groundwater. Gu et al. indicated that there were metal enrichments (Cd, Cu, Pd, and Zn) in the lower profiles of sludge-amended soil columns. Even under the greatest sludge application rate (150 g·kg^−1^), the proportions of the four heavy metals in the leachate over the experimental period were only 2.35%, 0.0453%, 0.244%, and 0.00889%, respectively [11]. Fang et al. carried out a leaching assessment to evaluate the potential leaching of heavy metals during composted sewage sludge application to soils. The authors pointed out that repetitive additions of compost favored the formation of reducing conditions due to the significantly increasing content of OM, causing accumulation in total contents for Cd, Cr, Cu, and Pb, but not enhancement in leaching concentrations [31]. Mortula indicated that both alum and lime treatment were capable of reducing leaching of heavy metals, while aluminum concentrations in the leachate increased with the increase in alum concentrations [32].

In addition to heavy metals, leaching of plant nutrients from sewage sludge might lead to pollution of groundwater. Therefore, in the process of sludge land use, we should not only pay attention to the migration of heavy metals, but also to that of plant nutrients. Li et al. found that the phosphorus content in soil after sludge application increased significantly in the range of 0–20 cm and there was no significant difference in the soil below 40 cm, which was related to the capacity of surface soil to accommodate phosphorus and the weaker mobility of phosphorus in the soil. Nitrogen in the form of nitrate nitrogen had a higher risk of leaching than phosphorus because it was difficult to be absorbed by soil particles [33]. Oladeji et al. suggested that the nitrate nitrogen content in groundwater increased in the remediation site of the mine where sludge was applied at 801 to 1815 Mg·ha^−1^ cumulative rate between 1972 and 2004 [34]. Mortula et al. found that alum treatment was capable of reducing phosphorus, copper, and ammonia leaching from sewage sludge, but increasing aluminum and chromium leaching [35]. Therefore, leaching experiments could also be used to rationally analyze the migration rules of plant nutrients after sludge land use, so as to comprehensively evaluate the environmental risks of sludge application.

In this paper, leaching characteristics of heavy metals and plant nutrients in the sewage sludge passivated by composite phosphorus-containing materials were studied in order to investigate the possible environmental impact of sludge land use. The contents were as follows: (1) to study the tread of pH and electricity conductivity (EC) in leachate, (2) to analyze the cumulative release of heavy metals and plant nutrients, and (3) to fit the cumulative release of heavy metals and plant nutrients with four models to choose the most suitable kinetic model of cumulative release. From the above aspects, the environmental behaviour of the immobilized sewage sludge was evaluated, which could provide a scientific basis for the land application of sewage sludge.

## 2. Materials and Methods

### 2.1. Collection and Preparation of Materials

The dewatered sludge was obtained from Changqing Municipal Wastewater Treatment Plant in the west of Ji’nan, in which the anaerobic/anoxic/aerobic process was adopted to treat urban sewage. The sludge was a mixture from the primary sedimentation tank and the secondary sedimentation tank. 0.3% polyacrlamide (PAM) was added for centrifugal dehydration and the solid content could reach 15–20%. The sludge was air-dried, homogenized, crushed by a grinder and passed through a 60 mesh sieve for further analysis. The background soil was taken from mature soil within 20 cm from the surface in the campus of Qilu University of Technology. Surface weeds, plant residues, and gravel were removed and then soil was air-dried, homogenized, crushed by a grinder, and passed through a 60 mesh sieve for properties analysis and a 2 mm mesh for column experiment.

Rock phosphate, whose main component was fluorapatite (Ca_5_(PO_4_)_3_F), was purchased from Taizhou Changpu Chemical Reagent Co., Ltd. It was a gray-white powder mainly composed of Ca(H_2_PO_4_)_2_·H_2_O, A small amount of CaSO_4_ and free phosphoric acid and was passed through a 60 mesh sieve for further analysis. Superphosphate was obtained from Sinopharm Chemical Reagent Co., Ltd.

The properties of sewage sludge, background soil and rock phosphate were shown in Table 1.

### 2.2. Experimental Methods

#### 2.2.1. Leaching Column Experiment

The test device was a customized leaching device consisting of a water pump, a flow meter, a leaching column, and a polyethylene collecting bottle. The leaching column was made of plexiglass cylinders with an inner diameter of 10 cm and an inner height of 100 cm. The end of the bottom was a porous flange connecting column, and the flange plate was arranged with a quantitative filter paper, a 100 mesh nylon filter, and a 2 cm thick quartz sand from bottom to top.

The soil column was composed of two parts. The lower part was hand-packed with immature soil into a height of 40 cm with a bulk density of 1.3 g/cm^3^. The immature soil, taken from more than 20 cm below the surface, was air-dried and sieved with a 2 mm mesh. It was filled into the column according to the method of Hou et al. [36].

The upper part was hand-packed with mixed matrix into a height of 20 cm. In the mixed matrix, the prepared sludge was mixed uniformly with background soil to obtain three treatments: control treatment (control), unstabilized sewage sludge treatment (USS), and stabilized sewage sludge treatment (SSS), whose components were shown in Table 2. In the SSS treatment, sewage sludge was mixed with rock phosphate and superphosphate in the proportion shown in Table 2. The mixture was maintained at 50% water content for 7 days at room temperature to ensure complete immobilization of heavy metals in sewage sludge, and was then air-dried, crushed by a grinder, and passed through a 60 mesh sieve. Three kinds of mixed matrixes were filled into the column according to the method of Hou et al. [36].

In order to ensure uniform distribution of deionized water added to the soil column, its top was covered with a 2 cm height of quartz sand and a 100 mesh nylon filter. The leachate was filtered through quartz sand, nylon filter, filter paper, and then flowed out from the bottom water collecting pipe into a polyethylene plastic bottle.

After the filling of the soil column was finished, the valve at the bottom of the soil column was opened and placed in a bucket filled with deionized water, making the soil wetted and saturated by the rising capillary water. The saturated condition was kept for 24 h, and then the column was freely drained to reach field capacity (30% on weight basis). Thereafter, the column was irrigated with 400 mL deionized water, and the leachate was collected into a polyethylene plastic bottle. When the same volume leachate was collected, it was considered that a single elution process was completed. Next 100 mL leachate was taken out, 5 mL (1 + 1) nitric acid was added, and it stored at <4 °C before the determination of Cu, Cr, and Zn. pH, EC, TN, TP, and TK of the leachate were measured after the elution process ended.

The elution process was carried out 12 times successively and the interval between two leaches was 12 h. The total volume of the used deionized water was 4.8 L, which was equivalent to the annual precipitation (rain) in the experimental area (Ji’nan, North of China). Ambient temperature throughout the experimental period was 12–16 °C.

#### 2.2.2. Analytical Methods

pH value and EC were determined in the 1:5 (*w*/*v*) suspension of solid sample and distilled water using a pH meter and a conductivity meter, respectively. OM, CEC, and moisture content were determined by the K_2_Cr_2_O_7_ volumetric method, the EDTA-ammonium acetate exchange method, and the gravimetric method, respectively. Total nitrogen (TN), total phosphorus (TP), and total potassium (TK) were measured by semi-micro Kjeldahl method, Mo-Sb colorimetric method, and alkaline fusion-flame photometric method, respectively [37].

Soil and sludge samples were first digested with HNO_3_-H_2_O_2_ [38], and then the contents of heavy metals (Cu, Cr and Zn) were measured by inductively coupled plasma optical emission spectroscopy (ICP-OES, Optima 2000DV, Perkin Elmer, Waltham, MA, USA).

All the used containers were soaked overnight with 20% (*v*/*v*) HNO_3_ in advance and then rinsed with ultrapure water. HNO_3_ and H_2_O_2_ were of guarantee grade and the rest reagents were of analytical grade. The ultrapure water was from a Millipore Milli Q system.

## 3. Results and Discussion

### 3.1. pH and EC in the Leachate

In the three treatments, pH values increased with the increase of leachate volume, and achieved their maximum values, which were 7.35, 7.57, and 7.60 in the control treatment, SSS treatment, and USS treatment, respectively, when the collected leachate reached 1.2 L (Figure 1). The increase of pH value was mainly due to rapid exchange between exchangeable base ions and H+. The addition of sludge increased exchangeable base ions, which was beneficial to the increase of pH. At the same time, ammonia could be produced by the deamination of OM in sludge. Thus, the pH values of leachate in the SSS and USS treatment were higher than those in the control treatment, which was inconsistent with the result of Gu et al. [11]. They indicated that application of sewage sludge induced a small temporary decrease in pH leachate, and attributed the decrease to production of organic acid from decomposition of OM. Increased pH values could result in an increase in the number of negatively charged surface sites in the soil, increasing the adsorption capacity of the soil for cationic metals and decreasing their mobility.

When the volume of leachate exceeded 2.8 L, pH values in the three treatments decreased to stable values. The final pH values were 7.31, 7.35, and 7.37, and no obvious difference was observed between the final pH values because that soil was a heterogeneous body with complex components and large buffer capacity [39].

EC values in leachates can reflect the total electrolyte activity in solution. EC values in the three treatments decreased rapidly when the leachate volume was less than 1.2–1.6 L. There was no significant difference among the three treatments, suggesting that at this stage, it was the exchangeable salt-based ions in soil, rather than those in sewage sludge, that entered into the leachate. The adsorption of salt-based ions by the lower soil column prevented the leaching of salt-based ions from sewage sludge (Figure 2).

In the control treatment, with the increase of leachate volume, EC values tended to be in a quasi-equilibrium state, indicating that the exchangeable base ions were depleted and other salt-based ions were gradually released into leachate. While in the USS and SSS treatment, their EC values increased significantly with application of sewage sludge due to enhancement of ions leaching into the soil solution. The highest EC values were observed, with values reaching up to 2210 μs/cm when the collected volume was 2.0 L in the SSS treatment and up to 2818 μs/cm when the collected volume was 1.6 L in the USS treatment. The results were similar to that of Penido et al. who demonstrated that the addition of sewage sludge represented an increase in EC values [40]. In the SSS treatment, the formation of phosphate precipitation with passivating agents such as rock phosphate and superphosphate made EC values lower than those in the USS treatment.

### 3.2. Leaching Characteristics of Heavy Metals

The leaching process of heavy metals is actually migration of metal ions between water, soil, and sewage sludge particles, which is mainly related to adsorption–desorption, complexation–dissociation, and precipitation–dissolution reactions. The exchangeable metal ions adsorbed on solid media are displaced or desorbed into aqueous solution and become free ions, which eventually follow the leaching solution out of the system. The concentration variations of Cu, Cr, and Zn with the volume of leachate were shown in Figure 3.

The release processes of Cu in the three treatments could be divided into two phases: rapid release phase and slow release phase. During the first phase. Cu concentration in the leachates decreased rapidly with the increasing volume of leachate. The released Cu increased in the following order: control treatment < USS < SSS, indicating that the sludge addition increased the Cu concentration in the leachate and the addition of composting phosphorus materials decreased the leaching Cu concentration. The immobilization of Cu was mainly due to the formation of the metal phosphate precipitates, surface complexation, and adsorption. The released Cu might be exchangeable Cu and ionic Cu. Its releasing rate was mainly determined by the migration rate of the leaching solution in soil column, which was related to the height of soil column, soil bulk density, and copper speciation in sewage sludge [41]. During the slow release phase, various kinds of stable Cu such as carbonate copper, organic copper, sulfide copper, and lattice copper could be released slowly under the continuous elution of water [42,43]. The final Cu concentrations in the three treatments maintained at 0 mg/L, 0.05 mg/L, and 0.05 mg/L, respectively.

The release processes of Cr in the control treatment were similar to those of Cu. While in the SSS and USS treatment, Cr leached out at a high rate so that Cr concentration did not change significantly at the beginning of leaching process. The concentrations of Cr decreased from 0.05–0.07 mg/L at the beginning to 0–0.01 mg/L in the end.

The land use of sewage sludge significantly increased the concentrations of Zn in the leachate. However, the Zn concentrations in the SSS treatment were higher than those in the USS treatment, indicating that the phosphate-bearing materials could not effectively reduce the mobility of Zn. Cao et al. pointed out that phosphorus-bearing materials show a limited effect on Zn immobilization and even had the potential of activating Zn [42]. The surface adsorption or complexation were primarily responsible for Cu and Zn immobilization. Flow calorimetry indicated that Cu adsorption onto rock phosphate was exothermic, while Zn sorption was endothermic [44,45]. In the SSS and USS treatment, there was more exchangeable Zn, leading to the slow release phase arriving later than in the control treatment.

The highest concentrations of Cu, Cr, and Zn in the leachates were 1.47 mg/L, 0.07 mg/L, and 1.49 mg/L, respectively, as shown in Table 3. These values were all below the limit values of the fourth level in the Chinese national standard for groundwater quality (GB/T14848-2017), indicating that it would not cause pollution of heavy metals to groundwater when the added sewage sludge was less than 10%.

Figure 4 showed the accumulative release of heavy metals with the increasing volume of collected leachate. The cumulative releasing processes of heavy metals could be divided into two stages. In the first stage, the cumulative release increased rapidly, which was due to the desorption of heavy metal ions from the surface of soil particles, and the more active forms of heavy metals entering the leaching solution at a faster speed. In the second stage, the cumulative release increased slowly and reached a state of equilibrium gradually. During the process, heavy metals adsorbing on the surface of soil particles decreased, and those in the micropores within the particles diffused slowly into the solution. The proportion of active heavy metals in this stage would also decrease.

Land use of sewage sludge significantly increased the accumulative release of Cu, Cr, and Zn. The releasing Cu and Cr in the SSS treatment were less than that in the USS treatment, while the opposite was true for the releasing Zn. The results showed that the use of rock phosphate and superphosphate decreased the migration of Cu and Cr, while increased that of Zn.

### 3.3. Release Kinetics of Heavy Metals

The immobilized sewage sludge enters into an open environment system by land use. In the open system, all chemical reactions occur in a dynamic state. It is more helpful for understanding the transformation and migration of elements in soil to study chemical kinetics. Their release process in soils is affected not only by the physical and chemical properties of soil, but also by interaction with other substances present in soil. Mathematical models used to analyze the geochemical behavior of elements in soils has become the focus of research. The commonly used dynamic models include first-order kinetic equation, modified Elovich equation, double-constant rate equation, parabolic diffusion equation and so on [46]. Their kinetics equations were shown as follows,

First-order kinetic equation:Ln*y* = a + b*x*(1)

Modified Elovich equation:*y* = a + bln*x*(2)

Double-constant rate equation:Ln*y* = a + bln*x*(3)

Parabolic diffusion equation:*y* = a + b*x*^0.5^(4)
where *y* was the cumulative release of heavy metals when the cumulative volume was *x*, and a and b were constants.

The releases of heavy metals in the leaching process were fitted by the first-order kinetic model, modified Elovich model, double-constant model, and hyperbolic diffusion model, as shown in Table 4. For the first-order dynamic equation, the square ranges of regression coefficient (R^2^) in the three treatments were 0.66–0.70 for Cu, 0.47–0.68 for Cr, and 0.55–0.74 for Zn, respectively. The three heavy metals could not be well fitted in the three treatments, which indicated that the leaching process of heavy metals was not explained fully by diffusion mechanism.

The modified Elovich equation is one of the most commonly used equations for describing the kinetics of heterogeneous chemisorption on solid surfaces. It is not suitable for a single reaction mechanism process but is very suitable for processes with large changes in activation energy during the reaction process [47]. The kinetic data in the three treatments fitted well with the modified Elovich equation, with their squares of regression coefficient of 0.96–0.98 for Cu, 0.85–0.99 for Cr, and 0.93–0.97 for Zn, respectively, showing that the migration of heavy metals was a complex heterogeneous dispersion process. The fitting correlation coefficient was relatively high for Cu and Zn, while it was low for Cr.

The double-constant model equation is actually a modified Frendlich equation, which is used to describe the heterogeneity of energy distribution and the different affinity of adsorption sites to heavy metals on the surface of soil particles. It is as applicable to complex systems as the modified Elovich equation. For the double-constant model equation, the square ranges of regression coefficient in the three treatments were 0.93–0.94 for Cu, 0.80–0.94 for Cr, and 0.86–0.96 for Zn, respectively.

For the hyperbolic diffusion model, the square ranges of regression coefficient in the three treatments were 0.88–0.92 for Cu, 0.69–0.95 for Cr, and 0.81–0.96 for Zn, respectively. The hyperbolic diffusion model was most suitable for describing the diffusion process of substances in particles. The poor fitting results showed that the internal diffusion of heavy metals was not the only limiting factor in the leaching process.

Comparing the fitting results of the four models, the order of the fitting degree was as follows: modified Elovich model > double-constant model > hyperbolic diffusion model > first-order kinetic model. It indicated that the release kinetics of heavy metals was not a simple first-order reaction, but a complex heterogeneous diffusion process controlled by precipitation and dissolution, adsorption and desorption, complexation and dissociation, etc. This result was consistent with Zheng et al. [41] and Zhang et al. [48]. They all concluded that heavy metals’ releasing processes in soil under simulated leaching conditions were not a simple first-order reaction, but a process of multifactor integrated control.

### 3.4. Leaching Characteristics of Plant Nutrients

Sewage sludge is rich in plant nutrients such as N, P, and K. Its land application would inevitably enhance these plant nutrients entering into subsoil and groundwater, which would lead to the pollution of groundwater. Thus, leaching characteristics of N, P, and K should also be studied in order to obtain overall environmental influence of sludge land use.

#### 3.4.1. Concentrations of TN, TP, and TK in Leachate

The concentration variations of TN, TP, and TK with the volume of collected leachate is shown in Figure 5. Sludge addition enhanced the leaching concentrations of TN, TP, and TK.

The concentrations of TN, TP, and TK in the control treatment decreased gradually and tended to be stable when the collected volume was more than 2.4 L, indicating that when the soluble plant nutrients finished leaching, the insoluble parts began to be released into the leachate. When the volume of collected leachate reached 4.0 L. the concentrations of TN, TP, and TK in the leachate were 1.00 mg/L, 0 mg/L, and 0 mg/L, respectively.

In the SSS and USS treatments, the leaching TN and TK decreased, and then increased to the highest concentration when the collected volume reached 1.6–2.4 L. In next stage, the concentrations decreased rapidly and then reached to stable levels when the collected volume reached 3.6 L. The stable TN and TK concentrations in the USS treatment were almost the same as to those in the SSS treatment, while they were both higher than those in the control treatment.

Unlike the leaching process of TN and TK, the concentration of leaching TP increased at the beginning of the leaching process. The concentrations of TP reached the highest values of 3.37 mg/L at 2.0 L in SSS treatment and 3.99 mg/L at 1.2 L in USS treatment, respectively. Lei et al. suggested that the higher EC was, the easier phosphorus was to be desorbed from the soil [49]. The higher EC in USS treatment prompted TP leaching, leading to reaching maximum of TP earlier. Then, TP concentration began to decrease and reached a stable state at 4.0 L. The stable concentration in the SSS treatment was about 1.0 mg/L higher than that in the USS treatment, because of the addition of composite phosphorus-bearing materials.

#### 3.4.2. Accumulative Release Characteristics of Plant Nutrients

Figure 6 shows the accumulative release of TN, TP, and TK in the leachate. The land use of sewage sludge significantly increased the cumulative release of TN, TP, and TK in the USS and SSS treatment. TN in the three treatments increased with the increasing volume of the collected leachate. There was no obvious difference among the three treatments when the collected volume was less than 1.6 L, suggesting that TN from sewage sludge was intercepted by the lower soil column. When the collected volume exceeded 1.6 L, the increasing rates of accumulative release of TN in the USS treatment and SSS treatment were much higher than that in the control treatment.

The accumulate release of TP in the control treatment did not increase significantly when the collected volume exceeded 1.6–2.0 L. The accumulative release of TP in the SSS treatment increased with the increasing collected volume, while in the USS treatment the increase slowed down when the collected volume exceeded 3.6 L.

The accumulative releases of TK in the USS treatment and SSS treatment were similar to TP in the USS treatment.

#### 3.4.3. Accumulative Release Model of Plant Nutrients

The release processes of the three nutrient elements in leaching process were fitted by four kinds of kinetics models. The equations and R^2^ were shown in Table 5. Analyzing the fitting results of the four models in the three treatments, the best fittings were modified Elovich model and double-constant model for N, modified Elovich model for P, and modified Elovich model and double-constant model for K, while the fitting of the first-order kinetics model was the worst. The fittings of USS and SSS treatment were better, and the fitting of control treatment was worse. The leaching of plant nutrients during the land use of sewage sludge was a complex heterogeneous diffusion process, which was similar to the leaching of heavy metals.

## 4. Conclusions

The addition of sewage sludge obviously changed the pH and EC of leachate. pH values in the SSS treatment and USS treatment were higher than that in the control treatment. The addition of sewage sludge increased cumulative release of heavy metals. The highest concentrations of Cu, Cr, and Zn in the leachates were all below the limit values of the fourth level in the Chinese national standard for groundwater quality (GB/T14848-2017). The cumulative releases of heavy metals increased rapidly at first and then slowly. The cumulative release of Cu and Cr in the SSS treatment was higher than that in the USS treatment, while Zn was in the opposite. The best fitting equation for cumulative leaching release of heavy metals was modified Elovich equation with their squares of regression coefficient of 0.96–0.98 for Cu, 0.85–0.99 for Cr, and 0.93–0.97 for Zn, respectively. Sludge addition also enhanced the leaching concentrations of TN, TP, and TK. The best fittings were the modified Elovich model and double-constant model for N and K, and the modified Elovich model for P. The leaching process of heavy metals and plant nutrients was a process of multifactor integrated control.

## Figures and Tables

**Figure 1 ijerph-16-05159-f001:**
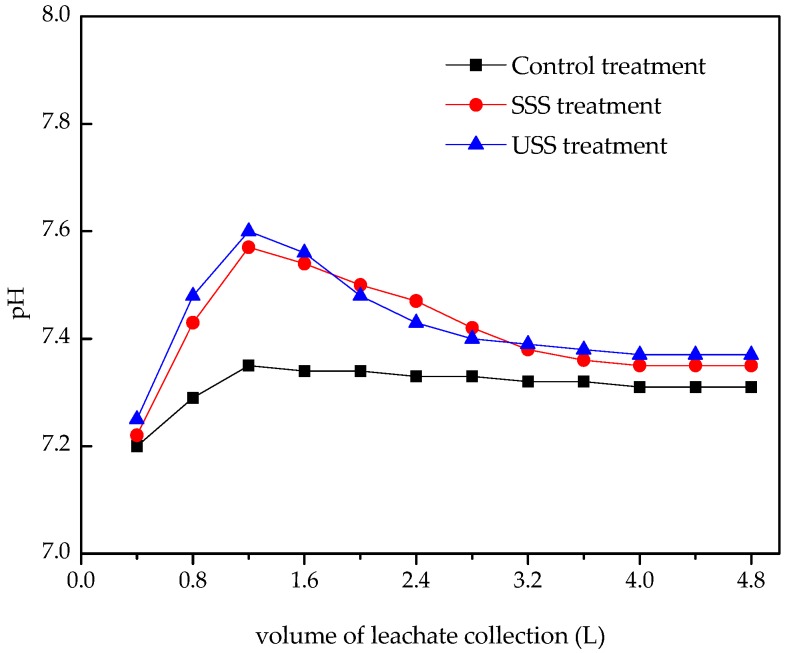
pH values in leachate collected from three treatments. USS: unstabilized sewage sludge treatment, SSS: stabilized sewage sludge treatment.

**Figure 2 ijerph-16-05159-f002:**
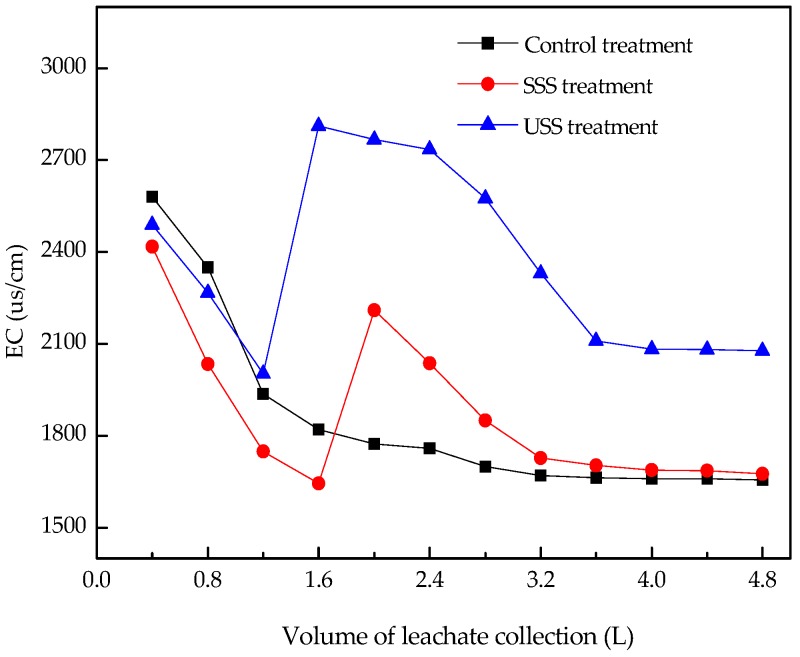
EC values in leachate collected from three treatments. EC: electrocity conductivity.

**Figure 3 ijerph-16-05159-f003:**
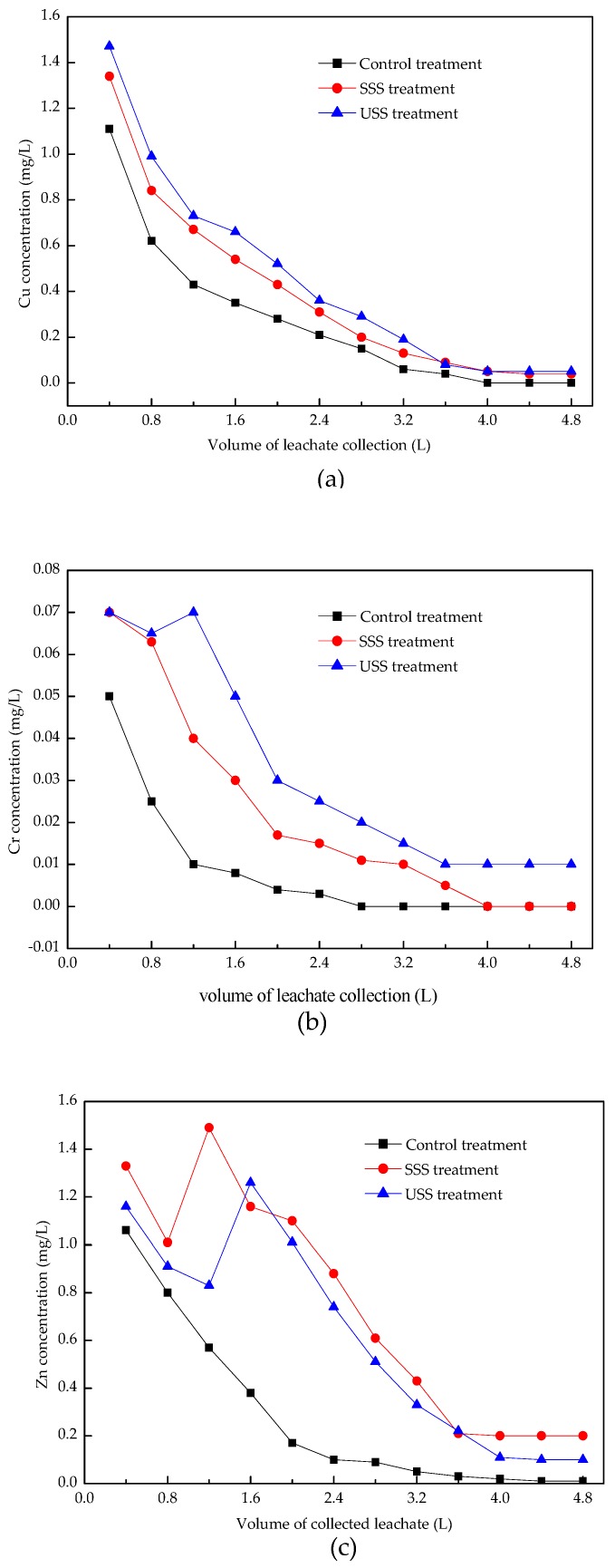
Concentration variations of Cu (**a**), Cr (**b**) and Zn (**c**) in leachate.

**Figure 4 ijerph-16-05159-f004:**
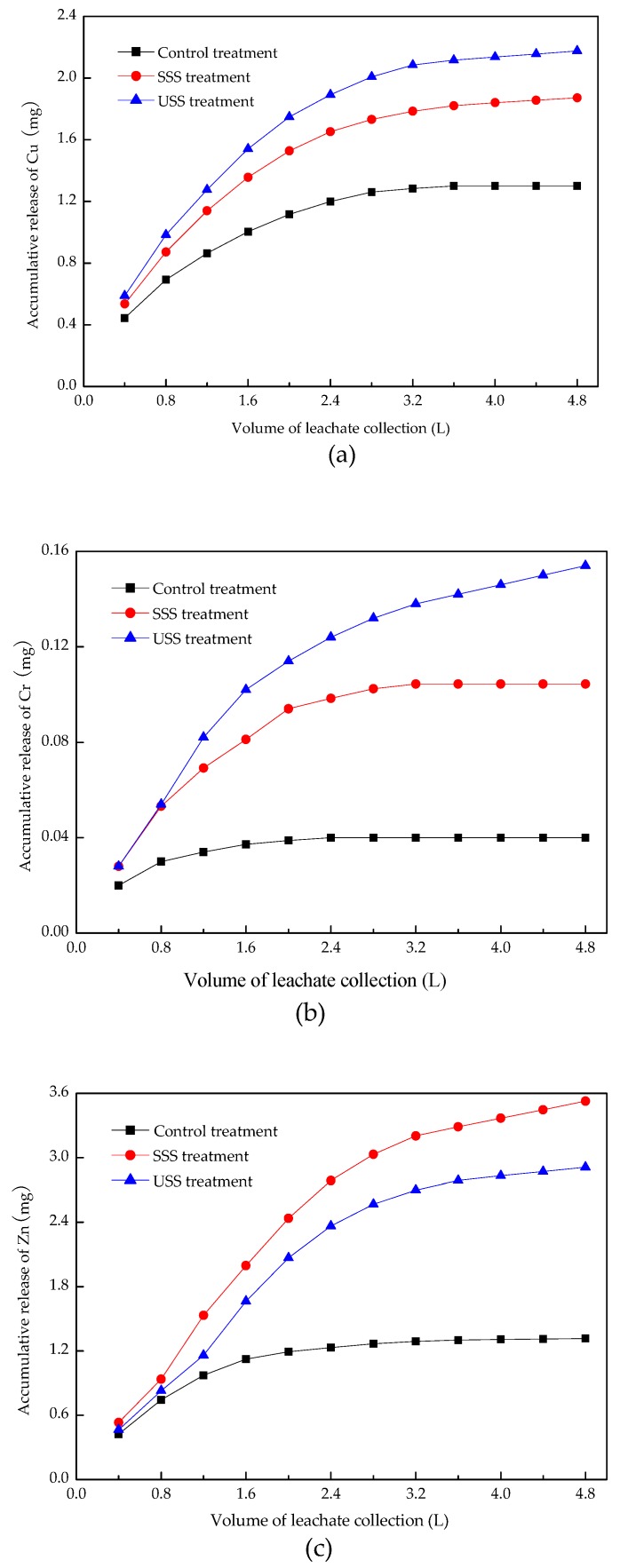
Accumulative release of Cu (**a**), Cr (**b**) and Zn (**c**) in leachate.

**Figure 5 ijerph-16-05159-f005:**
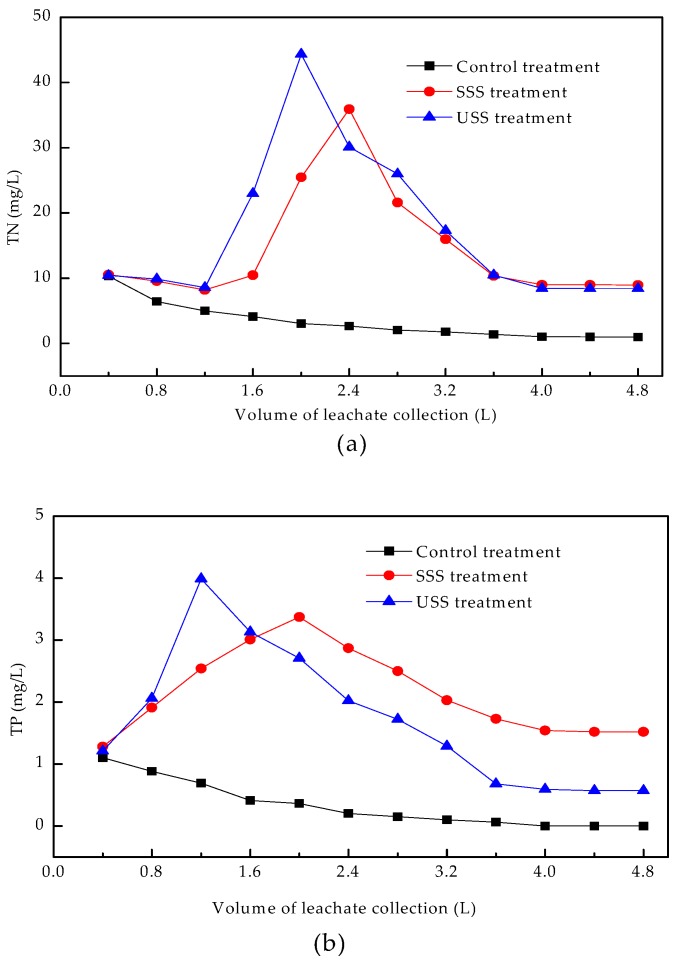
Concentrations of TN (**a**), TP (**b**), and TK (**c**) in leachate. TN: total nitrogen, TP: total phosphorus, TK: total potassium.

**Figure 6 ijerph-16-05159-f006:**
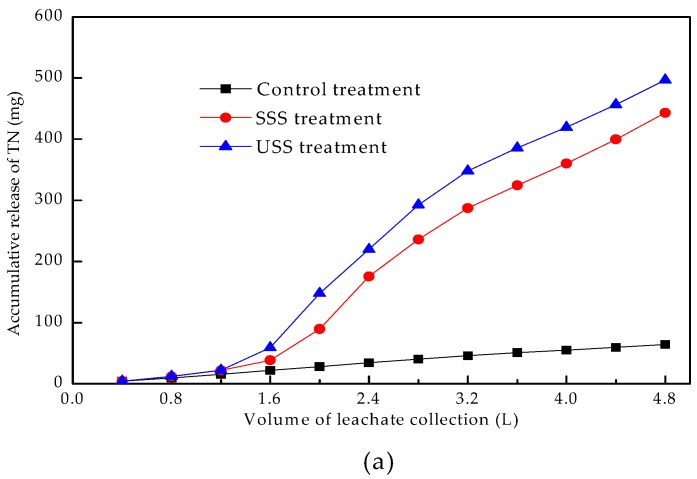
Accumulative release of TN (**a**), TP (**b**) and TK (**c**) in leachate.

**Table 1 ijerph-16-05159-t001:** Properties of the used sewage sludge, background soil, and rock phosphate.

Samples	Sewage Sludge	Background Soil	Rock Phosphate
pH	6.84 ± 0.10	6.86 ± 0.06	7.17 ± 0.13
EC (mS/cm)	1.88 ± 0.11	0.13 ± 0.004	0.40 ± 0.01
CEC (mmol/kg)	179 ± 3	27.3 ± 2.8	114 ± 2
OM (g/kg)	367 ± 2	9.76 ± 0.78	-
TN (g/kg)	17.5 ± 1.1	0.19 ± 0.16	-
TP (g/kg)	15.6 ± 0.8	0.38 ± 0.04	35.0 ± 1.2
TK (g/kg)	3.00 ± 0.60	2.98 ± 0.46	-
Cu (mg/kg)	75.5 ± 7.2	11.4 ± 1.4	45.3 ± 4.4
Cr (mg/kg)	402 ± 8	12.6 ± 2.0	13.8 ± 2.6
Zn (mg/kg)	2161 ± 68	74.2 ± 4.4	13.5 ± 1.2

“-” meant that the content was too low to be detected. EC: electricity conductivity, CEC: cationic exchange capacity, OM: organic matter, TN: total nitrogen, TP: total phosphorus, TK: total potassium.

**Table 2 ijerph-16-05159-t002:** The proportion (as %) of various substrates used in the three treatments.

Treatment	Background Soil	Sewage Sludge	Rock Phosphate	Superphosphate
Control	100	0	0	0
USS	90	10	0	0
SSS	90	8.85	0.885	0.265

Note: Each value represented on an air-dried weight base. USS: unstabilized sewage sludge treatment, SSS: stabilized sewage sludge treatment.

**Table 3 ijerph-16-05159-t003:** The highest concentrations of Cu, Cr, and Zn in the leachates.

Heavy Metals (mg/L)	Control Treatment	SSS Treatment	USS Treatment	Limit Value of the Fourth Level in GB/T14848-2017
Cu	1.11	1.34	1.47	<1.5
Cr	0.05	0.07	0.06	<0.1
Zn	1.06	1.49	1.26	<5

**Table 4 ijerph-16-05159-t004:** Accumulative release model of heavy metals.

Treatment		First-Order Kinetic Model		Modified Elovich Model		Double-Constant Model		Hyperbolic Diffusion Model	
		lny = a + bx	R^2^	y = alnx + b	R^2^	lny = alnx + b	R^2^	y = ax^0.5^ + b	R^2^
Cu	Control treatment	lny = 0.2381x − 0.1348	0.6964	y = 0.6879lnx − 1.2099	0.9811	lny = 0.5206lnx − 0.0941	0.9458	y = 1.0246x^0.5^ − 0.1459	0.9239
	SSS treatment	lny = 0.2263x − 0.2398	0.6889	y = 0.5749lnx − 1.0682	0.9801	lny = 0.4963lnx − 0.0233	0.9422	y = 0.8538x^0.5^ − 0.1828	0.9170
	USS treatment	lny = 0.1933x − 0.4609	0.6634	y = 0.3703lnx − 0.8112	0.9649	lny = 0.4282lnx − 0.2792	0.9289	y = 0.5447x^0.5^ − 0.2491	0.8840
Cr	Control treatment	lny = 0.2996x − 3.0445	0.6832	y = 0.5133lnx − 0.0740	0.9911	lny = 0.6581lnx − 2.7587	0.9378	y = 0.0796x^0.5^ − 0.0089	0.9470
	SSS treatment	lny = 0.2201x − 3.0824	0.6133	y = 0.0319lnx − 0.0621	0.9686	lny = 0.4969lnx − 2.8825	0.8986	y = 0.0468x^0.5^ − 0.0139	0.8788
	USS treatment	lny = 0.1032x − 3.5910	0.4747	y = 0.0075lnx − 0.0310	0.8502	lny = 0.2464lnx − 3.5074	0.7999	y = 0.0105x^0.5^ − 0.0204	0.6946
Zn	Control treatment	lny = 0.3624x − 0.1496	0.7339	y = 1.3790lnx − 1.4965	0.9746	lny = 0.7795lnx − 0.2087	0.9595	y = 2.0498x^0.5^ − 0.6526	0.9595
	SSS treatment	lny = 0.3593x − 0.3164	0.7420	y = 1.1286lnx − 1.2558	0.9635	lny = 0.7695lnx − 0.0413	0.9604	y = 1.7157^0.5^ − 0.5431	0.9489
	USS treatment	lny = 0.1803x − 0.3950	0.5509	y = 0.3573x − 0.8556	0.9281	lny = 0.4173lnx − 0.2388	0.8572	y = 0.5136x^0.5^ − 0.3316	0.8068

**Table 5 ijerph-16-05159-t005:** Accumulative release model of plant nutrients.

Treatment		First-Order Kinetic Model		Modified Elovich Model		Double-Constant Model		Hyperbolic Diffusion Model	
		ln*y* = a + b*x*	R^2^	*y* = aln*x* + b	R^2^	ln*y* = aln*x* + b	R^2^	*y* = a*x*^0.5^ + b	R^2^
N	Control treatment	ln*y* = 0.5391*x* + 1.9338	0.8546	*y* = 5.5703ln*x* − 0.5273	0.9956	ln*y* = 1.1104ln*x* + 2.5033	0.9959	*y* = 40.492*x*^0.5^ − 26.741	0.9871
	SSS treatment	ln*y* = 1.0225*x* + 1.9399	0.8810	*y* = 50.733ln*x* − 90.988	0.975	ln*y* = 2.0541ln*x* + 3.0591	0.9766	*y* = 317.14*x*^0.5^ − 289.54	0.9193
	USS treatment	ln*y* = 1.0319*x* + 2.0966	0.8422	*y* = 44.664ln*x* − 90.971	0.9717	ln*y* = 2.1158ln*x* + 3.194	0.9713	*y* = 363.69*x*^0.5^ − 321.87	0.9401
P	Control treatment	ln*y* = 0.2157*x*−0.3438	0.641	*y* = 0.4807ln*x* + 0.9528	0.9615	ln*y* = 0.4888ln*x* − 0.1494	0.9046	*y* = 0.7013*x*^0.5^ + 0.2319	0.8739
	SSS treatment	ln*y* = 0.5101*x* + 0.1428	0.8175	*y* = 4.328ln*x* + 2.6617	0.9317	ln*y* = 1.2274ln*x* + 0.5786	0.9863	*y* = 6.8223*x*^0.5^ − 4.6123	0.9885
	USS treatment	ln*y* = 0.5101*x* + 0.1428	0.7058	*y* = 3.4993ln*x* + 2.8453	0.9745	ln*y* = 1.1232ln*x* + 0.6274	0.9399	*y* = 5.3422*x*^0.5^ − 2.7681	0.9686
K	Control treatment	ln*y* = 0.1398*x* + 1.2722	0.5123	*y* = 1.3755ln*x* + 4.258	0.8732	ln*y* = 0.335ln*x* + 1.3848	0.8076	*y* = 1.9309*x*^0.5^ + 2.3121	0.7347
	SSS treatment	ln*y* = 0.4274*x* + 1.3633	0.7923	*y* = 8.4972ln*x* + 7.6749	0.9494	ln*y* = 0.9014ln*x* + 1.7992	0.9677	*y* = 13.067*x*^0.5^ + 6.1013	0.9586
	USS treatment	ln*y* = 0.4801*x* + 1.2437	0.8493	*y* = 9.8643*x* + 7.4019	0.9093	ln*y* = 0.9809ln*x* + 1.757	0.9735	*y* = 15.46*x*^0.5^ − 9.0399	0.9537

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
