# Peer review of "Leaching Characteristics of Heavy Metals and Plant Nutrients in the Sewage Sludge Immobilized by Composite Phosphorus-Bearing Materials"

_ijerph, 2019, doi:10.3390/ijerph16245159_

Round 1

Reviewer 1 Report

Manuscript Ref.: ijerph-659735

Title: Study on leaching characteristics of heavy metals and plant nutrients in the sewage sludge immobilized by composite phosphorus-bearing materials

This paper reports results and information on leaching of heavy metals and plant nutrients in the sewage sludge by using composite phosphorus-bearing materials. The work can be accepted after the following corrections and suggestions are taken into account.

Comments and specific suggestions:

The following suggestions and comments have been taken into account:

Paragraph 2.2.1: authors should specify which methods were used for determinations; even if they are common and everyday methods, the definition "standard laboratory procedures" alone means nothing. Table 1: the values with relative errors reported in the table are indicated without a logical criterion: for example 367.02±2.34 must be written 367±2. Paragraph 2.2.2. the authors use a 10cm x 100cm column. The column is filled with two different types of filling. Fill the column by capillarity and after 24 hours irrigate the column with 400ml. Have you considered a possible dilution? Can the paper filters at the bottom of the column adsorb metals or other substances which are then analytically determined? Paragraph 3.1: The authors show and well describe the probable mechanism that determines the variation of pH for the three treatments. However, I believe that, considering the type of experiment and the material used, variations in the order of 0.2 pH units are irrelevant or in any case indicatively at equilibrium.

Author Response

Point 1: Paragraph 2.2.1: authors should specify which methods were used for determinations; even if they are common and everyday methods, the definition "standard laboratory procedures" alone means nothing.

Response 1:The methods have been specified in line 154-157.

Point 2: Table 1: the values with relative errors reported in the table are indicated without a logical criterion: for example 367.02±2.34 must be written 367±2.

Response 2: The values in Table 1 have been modified.

Point 3: Paragraph 2.2.2. the authors use a 10cm x 100cm column. The column is filled with two different types of filling. Fill the column by capillarity and after 24 hours irrigate the column with 400ml. Have you considered a possible dilution? Can the paper filters at the bottom of the column adsorb metals or other substances which are then analytically determined?

Response 3: The three columns were drained to reach field capacity after keeping saturated for 24 hours. I think that the influence would not significant even if the dilution occurs at the beginning of the leaching.

    I'm sorry that we didn't consider the adsorption of heavy metals by the filter paper at the bottom of the column. Thank you very much for your comments. We will improve the experiment in the future. In addition, only one piece of filter paper was used in each column and its effect on adsorption was limit. What’s more, in this paper, we studied the leaching of heavy metals and nutrients in the three treatments. The adsorption existed in all the three treatments, and the comparative results wer also meaningful.

Point 4: Paragraph 3.1: The authors show and well describe the probable mechanism that determines the variation of pH for the three treatments. However, I believe that, considering the type of experiment and the material used, variations in the order of 0.2 pH units are irrelevant or in any case indicatively at equilibrium. 

Response 4: In this paper, the differences of pH value were 0.2-0.3 pH unites, which was small. The pH values of the leachates from the three treatments were determined at the same time, so there might be some differences between the pH values of the three treatments. Also, we gave a reasonable explanation according to the experimental results. However,your opinion was also reasonable, and we will find out the reason in our future research. Thank you for your opinion.  

Reviewer 2 Report

General Comments

The paper investigates an important aspect of chemical treatment of sewage sludge for reduction of leaching of heavy metals and nutrients. The authors did some literature review. However, there are some deficiencies that can be improved. I am putting couple of relevant papers below for consideration. As pointed by the literature review, there have been previous studies on similar concepts. However, the literature review doesn’t really specify the different materials used for immobilization of heavy metals and nutrients. Also, the authors failed to demonstrate the gap in knowledge they are trying to fill through this paper. The authors kept referring to three treatments. How can control be a treatment? pure background soil? Results can be presented better with enhanced focus on the role of phosphate based material on the immobilization. Conclusions are reasonable.

#Reduction of leachability of sewage sludge by alum treatment, April 2014, Journal of Environmental Engineering and Science 9(2):105-112

#Chemical Treatment of Sewage Sludge to Reduce Leachability, March 2012, Conference: Wastewater Purification and Reuse,: Heraklion, Greece

Specific Comments

Line 38, low cost Line 49, ..can contain toxic… Line 54, …the properties of soil Line 62, please consider rewriting the sentence. The use of “downward” here is confusing. Line 64-65, ratio shouldn’t be presented as percentage. Consider fixing it. Line 68, the use of the term “reducing condition” is not very clear. What condition the authors are referring to. Line 93-95, please specify the general treatment processes involved in the plant. Also specify if it is primary or secondary sludge or combined Line 100-103, I don’t see the point of making paragraph of one and half lines. Please consider merging them with other paragraph in the section. Also please explain the rationale for 20 cm or 60 mesh size. Why the amount of Zn, Cr, Cu (used in this study) in the background soil, SS and phosphate based material are not reported in the Table 1? It is quite relevant. Table 1 is a good fit for section 2.1 and section 2.2.1 should be the leaching column experiment and 2.2.2 should be the analytical methods. Characterization of existing sewage sludge and other materials are not part of the objective of this study (line 85-88). They are necessary and suitable for section describing the materials. There is no specific pattern of soil compaction. Why? If the compaction is followed by any standard procedure or any previous literature, please specify. There are more details necessary for the experimental setup, including porosity. The bulk density is specified. However, it is not clear if it includes the moisture or not. What is the rationale of the percentages used for Table 2? Please specify pH levels of the preserved samples. Why 5 mL nitric acid was used? Line 156-157, please specify the exact value of pH increase taking place. It doesn’t look that significant when reached stability. Line 186, please avoid “Our”, please use passive words. The results were consistent with … Figures should have a, b, c, rather than dumping three figures in one Under section 3.2, the authors specified the facts about changing concentrations of heavy metals. However, there aren’t much insight as to the reasons for increased concentrations while sludge and phosphate containing materials are used. It is specifically critical as the title of the paper uses “…immobilized..” It is not clear why the section 3.3 is created with accumulated data. It can be part of section 3.2. There is no new data. The same data is presented in a different scale. Line 254, get rid of the word “environmental”, Line 255, what is the meaning of “..has become the focus of research.” Is the modeling if the main focus of the study? Figure 5 and Figure 6 are identical. Figure 6 needs to be edited based on modified Elovich model. Figures 5-8 could have been presented similar to Table 4. I am not sure two identical presentation of data was done in the same paper in different ways. Line 324, eutrophication of groundwater? Really?

Author Response

Point 1: The paper investigates an important aspect of chemical treatment of sewage sludge for reduction of leaching of heavy metals and nutrients. The authors did some literature review. However, there are some deficiencies that can be improved. I am putting couple of relevant papers below for consideration. As pointed by the literature review, there have been previous studies on similar concepts. However, the literature review doesn’t really specify the different materials used for immobilization of heavy metals and nutrients. Also, the authors failed to demonstrate the gap in knowledge they are trying to fill through this paper.

#Reduction of leachability of sewage sludge by alum treatment, April 2014, Journal of Environmental Engineering and Science 9(2):105-112

#Chemical Treatment of Sewage Sludge to Reduce Leachability, March 2012, Conference: Wastewater Purification and Reuse,: Heraklion, Greece

Response 1: We have re-researched the relevant literatures and some new references have been added in the paper (reference 24-30, reference 32, reference 35). The mentioned papers have been cited as reference 32 and reference 35. The additives used for immobilization of heavy metals and nutrients were specified in line 58-60, line 70-72 and line 82-84. We demonstrated the gap in line 87-89.

Point 2: The authors kept referring to three treatments. How can control be a treatment? pure background soil? Results can be presented better with enhanced focus on the role of phosphate based material on the immobilization. Conclusions are reasonable.

Response 2: The upper part of the three leaching columns was hand-packed with different matrix. The matrix was shown in table 2. The rest of three treatments was same. We wanted to study the influence of land use of sludge and the addition of composite phosphorus-bearing materials on the leaching of heavy metals and plant nutrients. So three treatments in table 2 were set. The phosphate based material on the immobilization would be shown in another paper. In this paper, we also showed the immobilization effect of composite phosphorus-bearing in line 212-215, line 238-239.

 Specific Comments

Point 3: Line 38, low cost Line 49, ..can contain toxic… Line 54, …the properties of soil Line 62, please consider rewriting the sentence. The use of “downward” here is confusing.

Response 3: In line 38, line 49, line 54 and line 62, These sentences has been rewritten. The revisions have been highlighted in yellow.

The word “downward’in ine 62 has been deleted. The sentence has been rewritten in line 63-64.

Point 4: Line 64-65, ratio shouldn’t be presented as percentage. Consider fixing it.

Response 4: We have switched “ratios” into “proportions” .

Point 5: Line 68, the use of the term “reducing condition” is not very clear. What condition the authors are referring to.

Response 5: “Reducing condition” refers to the condition without oxygen. “Reduction condition”has the same meaning. In the conference [31], the author used “reducing condition”, and we chose to used this word in our paper.

Point 6: Line 93-95, please specify the general treatment processes involved in the plant. Also specify if it is primary or secondary sludge or combined Line 100-103, I don’t see the point of making paragraph of one and half lines. Please consider merging them with other paragraph in the section. Also please explain the rationale for 20 cm or 60 mesh size.

Response 6: The general treatment process of sewage was specified in line 98, and the sludge was also specified in line 99-100.

The paragraph, introducing the background soil, was merging with the first of paragraph of section 2.1.

The soil within 20cm from the surface is mature soil or mellow soil. It has high organic matter content and is suitable for plant growth. In this paper, we chose mature soil as the backgorund soil.

The treating methods of sewage sludge was chosen according to Chinese industry standard “ Determination method for municipal sludge in wastewater treatment plant (CJ/T221-2006)”. In the standard, the sludge sample was demanded to pass through 60-80 mesh size.

Point 7: Why the amount of Zn, Cr, Cu (used in this study) in the background soil, SS and phosphate based material are not reported in the Table 1? It is quite relevant.

Response 7: The values have been shown in Table 1.

Point 8: Table 1 is a good fit for section 2.1 and section 2.2.1 should be the leaching column experiment and 2.2.2 should be the analytical methods. Characterization of existing sewage sludge and other materials are not part of the objective of this study (line 85-88). They are necessary and suitable for section describing the materials.

Response 8: Table 1 has been moved to section 2.1. “Leaching column experiment” has been switched to section 2.2.1, and “analytical methods” has been switched to section 2.2.2.

Point 9: There is no specific pattern of soil compaction. Why? If the compaction is followed by any standard procedure or any previous literature, please specify. There are more details necessary for the experimental setup, including porosity. The bulk density is specified. However, it is not clear if it includes the moisture or not. What is the rationale of the percentages used for Table 2? Please specify pH levels of the preserved samples.

Response 9: The soil column was filled according to the reference [36]. which was shown in line 123-124, line 131-133,

The used soil and sewage sludge were air-dried and the proportions in Table 2 were represented on an air-dried weight base (line 139).

The rationale of percentage used in table 2 was obtained by our previous experiments. According to the especiation changes of heavy metals in sludge, we determined the addition amount of rock phosphate and superphosphate, and determined the ratio of sludge and soil according to the influence of sewage sludge on plant growth.

The pH values of the samples were shown in table 1.

Point 10: Why 5 mL nitric acid was used?

Response 10: Nitric acid was added to preserve the water samples and to prevent the precipitation of heavy metals. The heavy metals in the solution acidified by nitric acid could be stable for several weeks.

Point 11: Line 156-157, please specify the exact value of pH increase taking place. It doesn’t look that significant when reached stability.

Response 11: The exact pH values have been specified in line 168-170, and line 181.

Point 12: Line 186, please avoid “Our”, please use passive words. The results were consistent with …

Response 12: “Our result was ” has been modified into “The results were” in line 200.

Point 13: Figures should have a, b, c, rather than dumping three figures in one

Response 13: The multiple figures in Figure 3, Figure 4, Figure 5,and Figure 6 have been marked with (a),(b) and (c).

Point 14: Under section 3.2, the authors specified the facts about changing concentrations of heavy metals. However, there aren’t much insight as to the reasons for increased concentrations while sludge and phosphate containing materials are used. It is specifically critical as the title of the paper uses “…immobilized..”

Response 14: The influences of sewage sludge and phosphate containting materials were shown in line 218-222. line 233-234, line 236-237. 

Point 15: It is not clear why the section 3.3 is created with accumulated data. It can be part of section 3.2. There is no new data. The same data is presented in a different scale.

Response 15:The section 3.3 has been moved into section 3.2 and the title of section 3.2 has modified into “leaching characteristics of heavy metals”.

Point 16: Line 254, get rid of the word “environmental”,

Response 16: The word “environmental” has been deleted.

Point 17: Line 255, what is the meaning of “..has become the focus of research.” Is the modeling if the main focus of the study?

Response 17: The sentence was modified into ”That mathematical models were used to analyze the geochemical behavior of elements in soils has become the focus of research”.

Point 18: Figure 5 and Figure 6 are identical. Figure 6 needs to be edited based on modified Elovich model. Figures 5-8 could have been presented similar to Table 4. I am not sure two identical presentation of data was done in the same paper in different ways.

Response 18: Figure 5-8 have been presented in Table 5 (line 377).

Point 19: Line 324, eutrophication of groundwater? Really? 

Response 19: It has been modified into “pollution of groundwater”

Reviewer 3 Report

Very informative paper, well designed and presented, easy to understand and follow. It was a pleasure to read it.

As minor suggestions:

Maybe the authors can find a shorter and more focused title. I find it too tangled and I prefer shorter title, but it is not a must.

Table 2, 1st Column: please delete word "treatment" near Control, USS and SSS; you have already mentioned that is about Treatment in the head of the table. Also, considering that all sizes are expressed in%, I would remove the percentage symbol from the mathematical values (in the table) and introduce it in the title as follows: Table 2. The proportion (in%) of various substrates used in the three treatments.

Same observation fo Table 3, 1st column, where, in the head of the table should be Heavy metals (mg/L), and no unit of measure near each metal.

Author Response

Point 1: Maybe the authors can find a shorter and more focused title. I find it too tangled and I prefer shorter title, but it is not a must.

Response 1: the title has been modified into”Leaching Characteristics of Heavy Metals and Plant Nutrients in the Sewage Sludge Immobilized by Composite Phosphorus-bearing Materials”.

Point 2: Table 2, 1st Column: please delete word "treatment" near Control, USS and SSS; you have already mentioned that is about Treatment in the head of the table. Also, considering that all sizes are expressed in%, I would remove the percentage symbol from the mathematical values (in the table) and introduce it in the title as follows: Table 2. The proportion (in%) of various substrates used in the three treatments.

Response 2: “Treatment” near control, USS, SSS has been deleted in table 1.  

The percentage symbol from the mathematical values (in table 1) has been deleted.

The title was introduced as follows: Table 1. The proportion (in%) of various substrates used in the three treatments.

Point 3: Same observation fo Table 3, 1st column, where, in the head of the table should be Heavy metals (mg/L), and no unit of measure near each metal.

Response 3: The head of table 3 has been modified into”heavy metals (mg/L)”, and the units near each metal have been deleted.

Round 2

Reviewer 2 Report

It looks good on the most part. Quite improved from the previous version.